# Voltage-driven control of single-molecule keto-enol equilibrium in a two-terminal junction system

Chun Tang [1,2,5], Thijs Stuyver[3,4,5], Taige Lu[1,5], Junyang Liu [1,5], Yiling Ye[1], Tengyang Gao[1], Luchun Lin [1], Jueting Zheng [1], Wenqing Liu[1], Jia Shi[1], Sason Shaik [3] ✉, Haiping Xia [1,2] ✉ & Wenjing Hong [1] ✉

Keto-enol tautomerism, describing an equilibrium involving two tautomers with distinctive structures, provides a promising platform for modulating nanoscale charge transport. However, such equilibria are generally dominated by the keto form, while a high isomerization barrier limits the transformation to the enol form, suggesting a considerable challenge to control the tautomerism. Here, we achieve single-molecule control of a keto-enol equilibrium at room temperature by using a strategy that combines redox control and electric field modulation. Based on the control of charge injection in the single-molecule junction, we could access charged potential energy surfaces with opposite thermodynamic driving forces, i.e., exhibiting a preference for the conducting enol form, while the isomerization barrier is also significantly reduced. Thus, we could selectively obtain desired and stable tautomers, which leads to significant modulation of the single-molecule conductance. This work highlights the concept of single-molecule control of chemical reactions on more than one potential energy surface.

The control of chemical reactions at the single-molecule scale offers an opportunity for conceptually fabricating nanodevices by exploiting single-molecule reactions. Toward this goal, an ideal single-molecule reaction is expected to significantly modulate the electronic properties without involving significant changes in the molecular conformation. Tautomerization, describing the reversibly structural interconversion between a pair of well-defined tautomers with very similar molecular conformations, plays an essential role in chemistry[1] and biological system[2] and provides an ideal mode of operation to achieve the modulation of nanoscale electronic properties[3,4], which shows promising potential for the design of single-molecule devices[4–9]. However, the switchable tautomeric states at the single-molecule scale are usually observed to be in dynamic interconversion even under cryogenic conditions, since the pairs of tautomeric states are generally degenerate or quasi-degenerate with a low isomerization barrier. Some tautomerizations, however, have non-degenerate tautomeric states with the thermodynamic equilibrium strongly on one side, and the isomerization barrier to reach the other state is insurmountable (e.g., activation energies higher than 50 kcal mol$^{-1}$). While the reaction equilibrium and high barrier can support a robust tautomeric state, they also limit the molecule's ability to switch. Therefore, controlling the single-molecule tautomerization from the view of engineering the reaction barrier and thermodynamics driving force can be expected to provide an opportunity to reconcile the dilemma between switchability and robustness.

A potential energy surface (PES) determines both the kinetic barrier and the thermodynamic driving force of a chemical reaction,

[1]State Key Laboratory of Physical Chemistry of Solid Surfaces, College of Chemistry and Chemical Engineering, Xiamen University, Xiamen 361005, P. R. China. [2]Shenzhen Grubbs Institute, Department of Chemistry, Southern University of Science and Technology, Shenzhen 518055, P. R. China. [3]Institute of Chemistry, Edmond J. Safra Campus at Givat Ram, The Hebrew University, Jerusalem 91904, Israel. [4]Ecole Nationale Supérieure de Chimie de Paris, Université PSL, CNRS, Institute of Chemistry for Life and Health Sciences, 75 005 Paris, France. [5]These authors contributed equally: Chun Tang, Thijs Stuyver, Taige Lu, Junyang Liu. ✉e-mail: sason@yfaat.ch.huji.ac.il; xiahp@sustech.edu.cn; whong@xmu.edu.cn

directly associated with the reaction rate and preferential product in reaction equilibrium. In recent years, the reaction equilibrium has been demonstrated to show a dependence on the vicinal environment[9,10] and external force[11], suggesting a promising avenue to control tautomerization equilibria by modulating the PES. Additionally, oriented external electric fields (OEEFs)[12] have been used to reduce reaction barriers to accelerate chemical reactions, both in solution[12–18] and at the single-molecule scale[13,19]. Electric fields also play an important catalyzing role in many enzymatic processes[18,20]. Simultaneous control of both the thermodynamic driving force and the kinetic barrier, however, has not yet been well established at the single-molecule scale. Inspired by the control of the charge state of a single-molecule component through the application of a bias voltage[21], we were able in this work to access the PESs in both the neutral and charged states, providing an opportunity to explore the potential of thermodynamic and kinetic control in single-molecule reactions.

Specifically, we report here the control of a prototypical single-molecule keto-enol equilibrium with persistent tautomeric states at room temperature through modulation of an applied voltage by the scanning tunneling microscope break junction technique (STM-BJ)[22,23]. We find that the single-molecule operations involve two redox-related PESs, one corresponding to the uncharged state of the molecular bridge (MB; Fig. 1a) and the other one corresponding to a positively charged state of the MB (Fig. 1a). On the first PES (PES I, Fig. 1b), the keto form is the thermodynamically stable form with an unsurmountable barrier associated with a four-membered cyclic transition state (shown in the dashed frame of Fig. 1a)[24], suppressing dynamic and thermal interconversion into the enol form. Upon charge injection, the second PES is reached (PES II, Fig. 1b), showing a reversed thermodynamic driving force (the enol form is thermodynamically stable) and a dramatically reduced tautomerization barrier. Consequently, the keto form is readily converted into the enol form on PES II, and remains in this form until an electron is injected back into the MB from one of the contacts. By adjusting the voltage, we can reversibly operate the tautomeric states between an insulating keto form (σ bridge) and a conducting enol form (π bridge) that bridges the charge transport pathway of the two-terminal junction (Fig. 1c, d), leading to a remarkable conductance difference (amounting to a factor of ≈67) between tautomers. We also used the mechanically controlled break junction technique (MCBJ) based on a microfabrication chip to construct robust single-molecule devices with a fixed molecular

orientation. In the latter experimental setup, the alignment between the dipole moment of the molecular component and the electric field is observed to be critical for the electrically controlled tautomerization, i.e., the oriented electric field, resulting from the applied bias, acts as a non-negligible facilitator/inhibitor of the barrier crossing. This proposed control mechanism is further elucidated through theoretical calculations.

## Results

### Conductance measurements

To maximize the impact of the conductance difference between the tautomeric pairs, we placed the (enol-keto) tautomerization unit (red parts in Fig. 2a) in between two thioanisole anchors (blue parts in Fig. 2a), since this guarantees that the contact-to-contact transmission pathway passes through the tautomerization unit. The resulting MB can be synthesized in its keto-form **1** at the gram scale by an electrophilic substitution reaction in one step (see Supplementary Information). From the NMR characterization of the resulting compound (Supplementary Fig. 9), we were able to verify that it is indeed the keto form that dominates the tautomeric equilibrium in the solution. The methylene carbon with the sp³ hybridization present in **1** blocks the conjugation between the two sides of the MB. Consequently, **1** is expected to exhibit a significantly lower conductance in the sulfur-to-sulfur pathway than its corresponding tautomeric isomer **2**, i.e., the enol form with a π-bridge between the two phenyl groups. The enol form can be trapped by methylation, resulting in **2-OMe**, which was used as a reference in our experiments (Fig. 2a).

To investigate the charge transport of the designed single-molecule device, we dissolved 0.1 mM molecule **1** in 1,2,4-trichlorobenzene solvent and characterized the resulting solution by STM-BJ at room temperature. During a typical STM-BJ experiment, we continuously measure the conductance value while changing the distance between the gold tip and substrate electrodes. When a molecule bridges the gap between the two electrodes, the conductance change in response to changes in the distance becomes less pronounced, resulting in conductance plateaus in the conductance traces (conductance vs. stretching distance). As shown in Fig. 2e, the individual conductance traces characterized at a 0.1 V bias show conductance plateaus (blue traces) between $10^{-4}$ and $10^{-5}$ $G_0$ ($G_0$ is the quantum conductance, which equals 77.5 μS)[25]. After increasing the bias to a moderate 0.6 V, we observe another plateau with a higher conductance value between $10^{-2}$ and $10^{-3}$ $G_0$ (red traces).

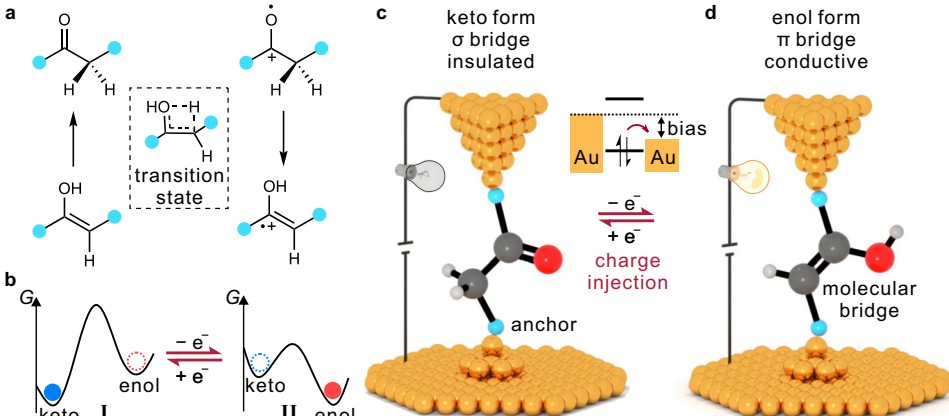

**Fig. 1 | Schematic diagram of a nanoscale device based on single-molecule tautomerization. a** Reaction scheme of the keto-enol tautomerization via its four-membered transition state. The four-membered transition state is shown inside the dashed box. The cyan cycles represent anchor groups. **b** The sketched PESs I and II in different charged states are correspondingly shown. The blue and red cycles represent keto and enol forms. The solid and dash cycles represent reachable and unreachable states. **c**, **d** Schematic representation of the STM-BJ setup and the

tautomerization-based nanoscale device. The device is in the low-conductance keto state in which a σ bridge connects the two contacts (**c**). The charge injection leads to the transformation to the high-conductance state with a π bridge connecting the two electrodes (**d**). The sketched energy diagram of the charge injection is shown in the middle. The cyan balls represent anchor groups. The gray, red, and white balls represent carbon, oxygen, and hydrogen, respectively. The brownish-red arrows represent the charge injection process.

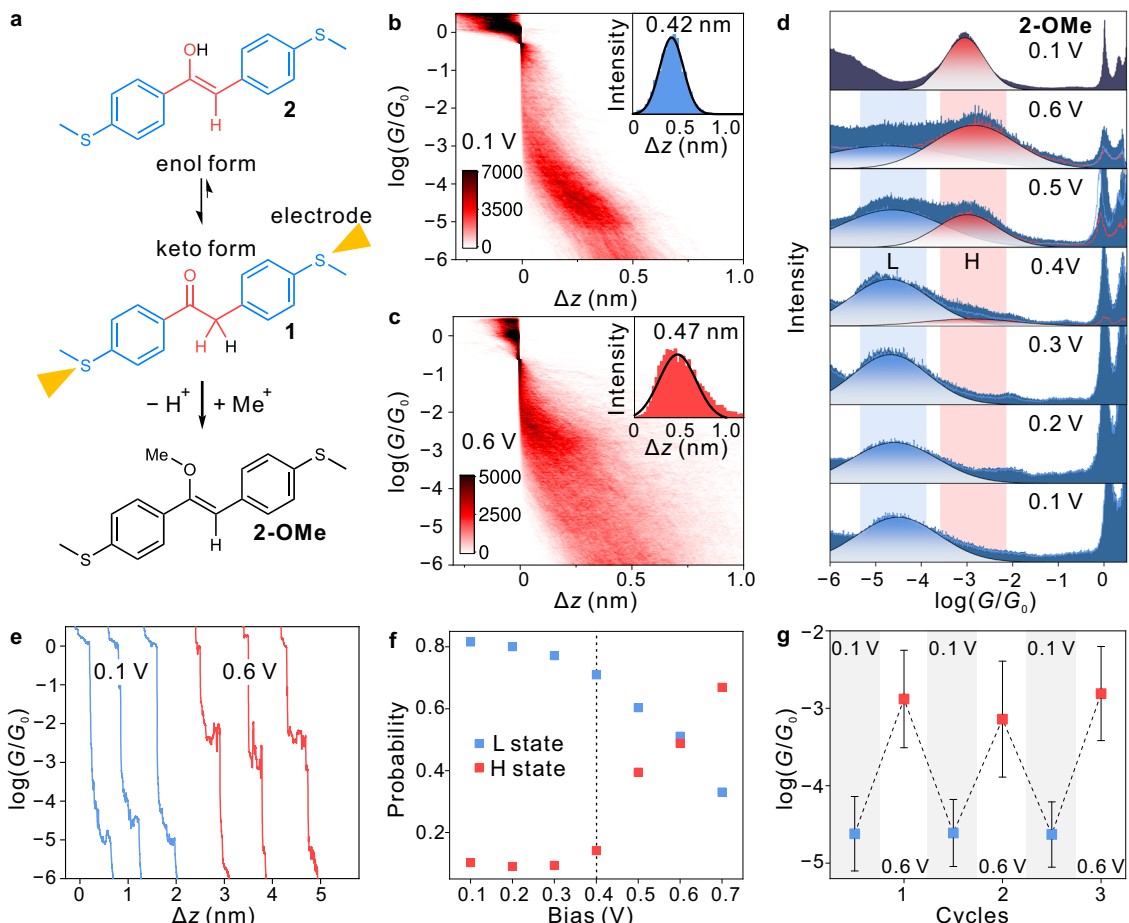

**Fig. 2 | Single-molecule conductance characterization. a** Chemical structure of compound **1**, which is formally in a tautomeric equilibrium with its enol isomer **2**. The staggered equilibrium arrows represent that the keto form dominates this equilibrium. Compound **2-OMe**, the methylated analog of enol form **2**, is formed by methylation of compound **2**. **b, c** 2D conductance histograms of compound **1** at 0.1 V (**b**) and 0.6 V (**c**) bias, with the corresponding stretching distances shown in the insets with the blue (**b**) and red (**c**) histograms. The color scales represent the intensity. **d** 1D conductance histograms of compound **1** characterized at different biases ranging from 0.1 to 0.6 V. The low and high-conductance states are labeled 'L' and 'H', respectively. The 1D conductance histogram of compound **2-OMe** at 0.1 V bias is shown on the upper side. The blue and red lines originate from the 1D conductance histograms of the traces of 'L' and 'H', respectively, with the blue and red areas obtained through Gaussian fitting. The blue and red areas represent the 'L' and 'H' states, respectively. **e** Typical individual conductance traces of compound **1** characterized at 0.1 V (blue lines) and 0.6 V (red lines) biases. **f** Distribution probabilities of states 'L' and 'H' plotted against different biases for a constant tip speed (5 nm s⁻¹). The dashed black line represents the threshold, after which the 'H' state becomes observable. **g** Peak centers of the dominant conductance peaks of the corresponding 1D conductance histograms with alternation of the bias between 0.1 and 0.6 V. The error bars correspond to the standard deviation of the Gaussian fitting. The blue and red colors represent the 'L' and 'H' states, respectively.

By overlapping more than 3000 such conductance-distance traces, we obtain the two-dimensional (2D) conductance histograms shown in Fig. 2b, c (Supplementary Fig. 3). These 2D histograms reveal similar shapes of conductance plateaus in distinct conductance ranges. We further analyze the stretching distance distributions of the conductance plateaus. As shown in the insets of Fig. 2b, c, the stretching distances of the conductance plateaus are similar at biases of 0.1 and 0.6 V. Note that there is a 0.5 nm snap-back distance after the rupture of the gold-gold atomic contact[26]. By adding these snap-back distances to the stretching distances, the calibrated junction lengths of 0.92 and 0.95 nm are obtained at 0.1 and 0.6 V, respectively. The consistency of these stretching distances—in reasonable agreement with the expected molecular length—suggests that a robust sulfur-to-sulfur charge transport pathway is kept at different bias voltages. The robustness of the sulfur-to-sulfur charge transport pathway was also confirmed in a control experiment with one of the −SMe anchors of **1** removed, which results in no explicit molecular signal in a wide bias range (Supplementary Fig. 17). As such, this control experiment rules out the alternative hypothesis that the high-conductance state results from the contact migration, e.g., from one of the sulfur contacts to the oxygen site.

One-dimensional (1D) conductance histograms were constructed without data selection at different bias voltages up to 0.6 V. As shown in Fig. 2d, the peak center (determined by Gaussian fitting) of the 1D conductance histograms at 0.6 V is located at $10^{-2.82}$ $G_0$, which is a 67-fold increase relative to the peak at 0.1 V ($10^{-4.65}$ $G_0$). The 1D conductance histograms at biases ranging from 0.1 to 0.3 V are all very similar, showing an approximately invariant single conductance peak centered around $10^{-4.6}$ $G_0$, which we defined as the low-conductance state and labeled 'L'. Increasing the bias beyond 0.4 V, we observed a double peak distribution, with a new peak gradually emerging at around $10^{-2.8}$ $G_0$, which we defined as the high-conductance state and labeled 'H'. The 'H' state became increasingly prominent as the bias increased and started to dominate at the bias voltage of 0.6 V. This dramatic conductance change suggests that the applied bias effectively transforms the nonconductive **1** to a highly conductive state similar to **2**. To confirm the nature of this transformation, the stabilized enol form **2**, i.e., the methylated derivate **2-OMe** (Fig. 2a), was synthesized and characterized at 0.1 V (Fig. 2d). The single conductance peak of **2-OMe** is centered at $10^{-3.06}$ $G_0$ (Fig. 2d). Increasing the bias up to 0.6 V does not discernably change the location of this peak

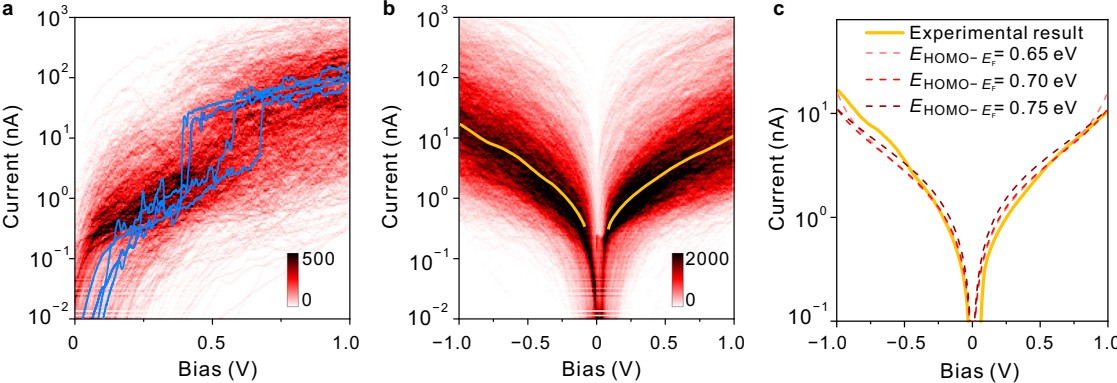

**Fig. 3 | I-V characterization. a** 2D I-V histogram constructed from the 831 traces exhibiting current jumps. Four representative I-V curves, showing the transformation from a low-conductance state to a high-conductance state, are highlighted in blue. **b** 2D I-V histogram constructed from 2542 (out of 3234) traces. The orange line was achieved by the Gaussian fitting in each row of bins (see details in the "Methods" section). The color scales of **a**, **b** represent intensity. **c** The orange line is the experimental fitted I-V curve from (**b**). The three dash lines are theoretical Lorentzian-fitted I-V curves for which the HOMO-Fermi level alignments ($\Delta E_{\text{HOMO}-E_{\text{F}}}$) alignments are varied.

(Supplementary Fig. 4). Meanwhile, the conductance plateau of **2-OMe** (Supplementary Fig. 4) is slightly longer than the 'L' state and is similar to the 'H' state (Supplementary Fig. 27), suggesting that the 'H' state has a similar junction geometry to **2-OMe** within the bias ranging from 0.1 to 0.6 V. The similarity in the conductance characteristics for **2-OMe** and the 'H' state suggests that the emergence of the 'H' state at high bias is associated with a conversion from the keto to the enol form. Because of the thermodynamic preference for the keto-form (combined with the high kinetic barrier) on the neutral PES (vide infra), we propose that the 'H' state is the positively charged enol form of **2**. As a control experiment, we also performed the electrochemically gated[27] STM-BJ experiment on **1** in a four-electrode system (Supplementary Fig. 15), proving that the conductance state after one-electron oxidation of **1** is similar to the 'H' state achieved at a higher bias (Fig. 2d). If there is a transformation from keto to enol form, we would expect a change of band gap since the enol form is a more conjugated structure. As characterized by the UV-Vis absorption (Supplementary Fig. 16), we found that the one-electron oxidized species of **1** shows a red-shifted absorption similar to **2-OMe**, reinforcing our hypothesis that the high-conductance state should be the one-electron oxidized enol form.

Through the analysis of the occurring probabilities of the 'H' and 'L' states at different bias voltages, we constructed the scatter diagram shown in Fig. 2f. The probability of the 'H' state begins to rise when the bias exceeds 0.4 V, while the probability of the 'L' state decreases correspondingly. The two probabilities become almost equal at 0.6 V bias. At a 0.7 V bias, the probability of the 'H' state further increases to 67%. At voltages exceeding 0.7 V, the formed single-molecule junctions become increasingly unstable, which renders characterization of the corresponding probabilities unfeasible. More importantly, we find that the transformation between the 'H' and 'L' states is reversible. To assert this, we changed the bias voltages between 0.1 and 0.6 V alternatingly. We collected approximately one thousand conductance traces for three subsequent cycles to determine the most probable conductance distributions (Fig. 2g and Supplementary Fig. 5). The resulting plot clearly demonstrates the robust and reversible operation of the single-molecule device.

To further understand the transformation between the 'H' and 'L' states, we also recorded the evolution of the current as a function of continuously varied bias for individual junctions (details are described in the "Methods" section)[28]. As shown in Fig. 3a, 831 out of the 13,850 so-recorded I-V curves exhibited clear current jumps without fixed threshold bias (four representative I-V traces are shown in Fig. 3a). It is worth noting that the current jumps happened in both positive and negative bias. This observation is consistent with the finding that the

'H' states become more dominant as the magnitude of the bias increases, regardless of whether the bias is positive (Fig. 2) or negative (Supplementary Fig. 6). As such, we used the absolute bias value to construct the 2D I-V histogram for clarity (Fig. 3a). The other traces exhibit a regular proportional increase in the current with increasing bias. From the I-V curves without a current jump (Fig. 3b), we determined the approximate alignment of the Fermi level with respect to the main eigenchannel, i.e., the HOMO of **1**, through a Lorentzian fitting approach (see Supplementary Note 4 for a detailed methodology[29]). An optimal agreement with the experimental data is obtained for an approximate energy difference between the Fermi level and the HOMO channel of 0.65–0.75 eV (cf. Fig. 3c). The obtained alignment—which indicates that the fully resonant transport regime is not reached for the bias window probed in our experiments—agrees within 0.2–0.3 eV with the location of the HOMO-peak in the calculated transmission spectrum for a junction containing **1** (cf. Fig. 5c). Furthermore, one can observe from Fig. 5c that within the scanned bias window, the transmission probability for **1** remains more or less constant. This finding explains why the 'L' state in Fig. 2d remains fairly robust, and provides further evidence that the sudden emergence of the 'H' state at voltages exceeding 0.4 V is caused by a structural modification of the MB, i.e., by tautomerization. It should also be noted that the ratio in the transmission probabilities between this low-conductance state and the hypothetical high-conductance state, i.e., the positively charged enol-species (vide infra), agrees reasonably with the experimentally determined conductance ratio (≈100:1) within the bias window.

To further explore the potential of using the molecular component to design a molecular device, we used the MCBJ technique (upper panel, Fig. 4a) with microfabrication chips (Supplementary Fig. 29) to construct robust single-molecule junctions to understand the influence of molecular orientation. Using the microfabrication chips, we can fix the orientation of molecule **1** with a long lifetime of up to several minutes (Supplementary Fig. 31). In Fig. 4b, the I-V curves for a typical single-molecule junction formed in this way are shown. Transition voltage spectroscopy (TVS)[30] was utilized to determine the transition voltages in the negative and positive bias regions, which were found to be at 0.38 and 0.81 V, respectively (Fig. 4c). An unequivocal dependence on the orientation of the bias/electric field can be observed for this junction: at a negative bias, clear transformation behavior is obtained, whereas the current changes fairly linearly at a positive bias (inset, Fig. 4b). Obviously, the initial assembly of the molecular device happens in random orientation (bottom panel, Fig. 4a), and accordingly, we can observe both orientations in different

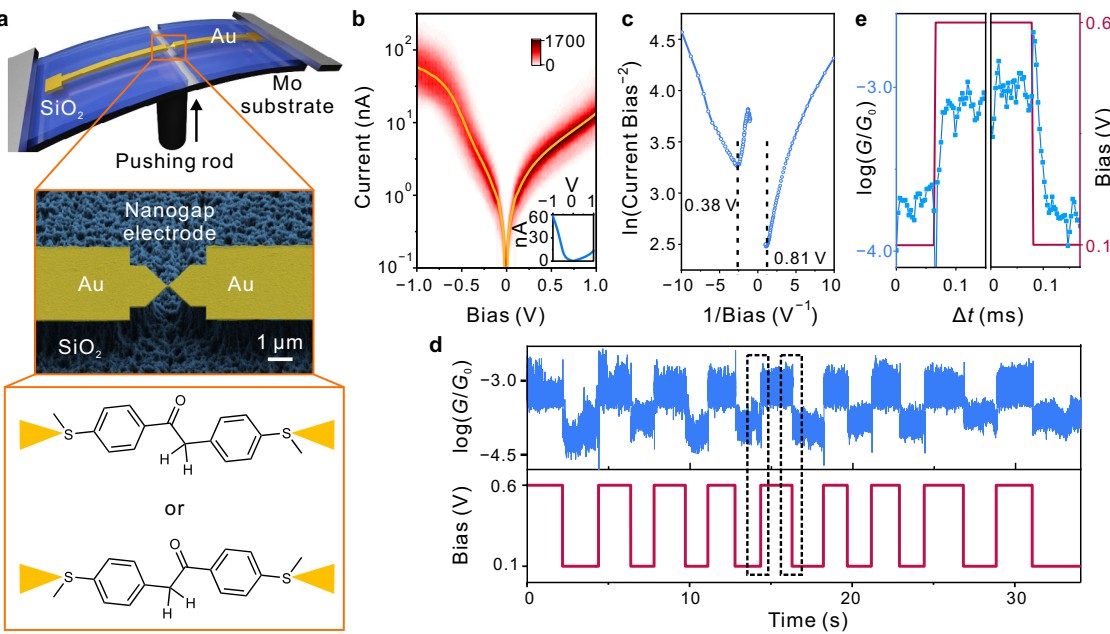

**Fig. 4 | Constructing single-molecule devices based on MCBJ chip. a** Schematic of the MCBJ chip. The SEM image of the nanogap electrodes is shown with a 1 μm scale bar. Molecular component **1** is expected to have a random orientation between the nanogap electrode. **b** 2D I-V histogram constructed from 1445 (out of 2302) traces exhibiting current jumps. The representative I-V curve is shown in the yellow line, which is replotted in the inset figure (current vs. bias) with the current value on a linear scale. The color scale represents intensity. **c** Based on the representative I-V curve, the TVS is shown correspondingly for both the negative and positive bias. The absolute values of transition voltage are shown correspondingly. The unit of the x-axis is V⁻¹. **d** The conductance trace of a single-molecule device shows the switching behavior with the change of bias. The conductance and bias values are shown in blue and brownish-red colors. **e** The data within the dashed frames of (**d**) is zoomed to show the switching with bias changed between 0.1 and 0.6 V.

long-lifetime junctions, a negative (e.g., Fig. 4b) and positive (e.g., Fig. 4e, also see Supplementary Fig. 30) bias trigger the transition.

At the same time, the transformation behavior is very robust during bias change between 0.1 and 0.6 V (Fig. 4d). We also tried to characterize the transformation speed as shown in Fig. 4e, and found that the transformation speed should be faster than 0.1 ms in both the keto-to-enol and enol-to-keto processes. It should be noted at this point that the coupling of a chemical reaction into the single-molecule device leads to an electronic property that is not analogous to that of a traditional electronic device. In a conventional molecular junction, the transition voltage corresponds to a crossing of the HOMO or LUMO transport channel and is also associated with a non-linear increase in current. Here, the transition voltage also corresponds to the state change in the single-molecule keto-enol tautomerization, which also exhibits good reversibility. Importantly, the orientation dependence and fast transformation speed established above provide the device with rectifying properties (i.e., polarized electronic behavior), which could lead to various interesting applications.

## DFT calculations

To further elucidate the controlling mechanism, we performed DFT calculations on a model system consisting of bare molecule **1** in the absence of Au clusters (Fig. 5a). First, the PES associated with the neutral species was considered. In accordance with the experimental observations, we observe that the enol form lies 9.3 kcal mol⁻¹ higher in energy than the keto form (see "Methods" Section for computational details). This energy difference persists even in the presence of an electric field corresponding to the field exerted by the approximate threshold voltage for manipulating in the junction ($F_Z$ = +0.05 V Å⁻¹; cf. Fig. 5a). This finding constitutes an unequivocal confirmation that tautomeric interconversion cannot occur on the uncharged PES (in blue).

Concerning the kinetics of the tautomeric interconversion process, our calculations indicate that the barrier separating the keto and enol forms is truly unsurmountable; the direct tautomerization mechanism, i.e., the H-atom shuttling from the carbon to the oxygen center (Fig. 5a), corresponds to a potential energy hill of 59.3 kcal mol⁻¹. We considered alternative mechanisms involving H₂O as well[15,31,32]. While our calculations suggest that the presence of H₂O can reduce the kinetic barrier, while the thermodynamic driving force cannot be reversed, so the tautomerization remains unfeasible (Supplementary Fig. 8). We performed control experiments in the glovebox (Supplementary Fig. 2a), and we also humidified the solvent environments (Supplementary Fig. 2b–e). Neither of these modifications to the reaction conditions changed the experimental results significantly, suggesting that H₂O does not significantly accelerate or inhibit the tautomerization process in the STM-BJ experiments. The isotope effect is also proven experimentally to be weak (Supplementary Fig. 1), which rules out a proton tunneling mechanism. Furthermore, no other acid or base source is present in the reaction medium to catalyze the process. Finally, since the experiment is performed in the absence of a light source, any mechanism involving a transition—upon absorption of a photon—to the PES of an (equally uncharged) excited state can also be ruled out[33–35].

Our calculations suggest that removing an electron from the neutral model system dramatically impacts the thermodynamics driving force, as well as the kinetics associated with the keto-enol tautomerization. In the positively charged system (Fig. 5a in red), the enol form becomes the thermodynamically stable form, with a driving force of 7.2 kcal mol⁻¹. This thermodynamic reversal from keto to enol preference upon oxidation of tautomeric compounds is well documented in the literature[36–39]. Removal of an electron significantly also reduces the energy barrier separating the two states: the barrier associated with this reaction pathway is almost cut in half from 59.3 to 35.8 kcal mol⁻¹ for the cationic species with +0.05 V Å⁻¹ electric field applied with the orientation of blue arrow shown in Fig. 5a. If the direction of the electric field is reversed (i.e., −0.05 V Å⁻¹ with the orientation of orange arrow, and Supplementary Fig. 6), we can

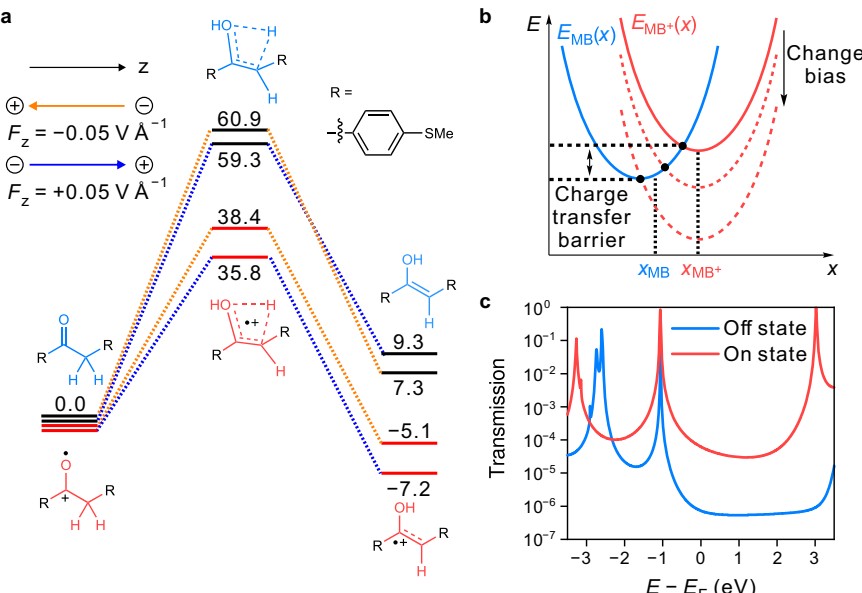

**Fig. 5 | Calculated reaction profile and transmission coefficient. a** Reaction profile (in kcal mol$^{-1}$) associated with the tautomerization reaction on the uncharged PES (black), and on the PES of the radical, cationic species obtained after charge injection (red), calculated at B3LYP/def2-TZVP level of theory. An electric field corresponding to the field exerted by the approximate threshold voltage in the junction has been included in different directions (shown in orange and blue arrows). The PES in the different directions of the electric field is linked by orange (−0.05 V Å$^{-1}$) and blue (+0.05 V Å$^{-1}$) dashed lines. **b** Schematic representation of the Marcus parabolas, describing the energy as a function of the environmental dielectric coordinate $x$, for the uncharged MB species (parabola in blue centered around $x_{MB}$) and the charged MB (MB$^+$) species (parabola in red centered around $x_{MB^+}$). As the molecule-metal potential difference changes, the MB$^+$-parabola shifts vertically. **c** Calculated transmission coefficients (dimensionless) of the low- and high-conductance states. The blue and red curves represent low- and high-conductance states.

observe a relative increase in the barrier height for the cationic species of several kcal mol$^{-1}$, suggesting that the OEEF is responsible for the polarized electronic behavior observed in the microfabrication chip setup. A qualitative valence bond analysis reveals the root cause of the dramatic change in the reaction profile upon charge injection (see Supplementary Note 2 for details). While the calculated barrier for the charged model system is still relatively high, one can expect junction effects, such as local heating[3], mechanical strain, and hybridization with the Au-contacts[40], to further assist in the catalysis of this process.

At this point, it is important to note that our experimental results indicate that the charging event does not require a crossing between the chemical potential of one of the contacts and the transport channel associated with the HOMO of the MB, cf. the Lorentzian fitting in Fig. 3c. Furthermore, Fig. 2f indicates no fixed threshold bias triggering the transformation; beyond a bias of 0.4 V, the proportion of 'H' traces gradually increases. These observations suggest that the charge injection triggering the tautomerization reaction is a thermal transition (cf. the Marcus model of electron transfer[41,42]); as the bias is increased, the charged and uncharged PESs approach, so that at some point, cross-over from one to the other becomes possible (Fig. 5b).

Support for the latter hypothesis can be found in the exact voltage range at which the current jump starts to occur: at a bias of 0.4 V, the Fermi level can be estimated to be 0.4–0.5 eV, or approximately 10 kcal mol$^{-1}$, away from the HOMO channel, which reasonably matches the thermodynamic preference for the enol-form over the keto-form on the charged PES. As such, starting from this voltage value, the energy required to oxidize the bridge is effectively compensated by the energy released through tautomerization on the charged surface, so that an overall transition from the uncharged, insulating keto-form to the charged, conducting enol-form becomes (thermodynamically) favorable. As an extra control experiment to confirm that the process is indeed thermal in nature, we performed the STM-BJ experiments

under a 0.5 V bias at different tip speeds, which resulted in varying lifetimes of the junctions (Supplementary Fig. 18). We clearly observed that as the average lifetime of the junctions increases, the conversion ratio from 'L' to 'H' increases accordingly. As such, the longer the molecular bridge is exposed to the triggering voltage, the higher the probability that the (thermal) barrier can be crossed.

## Discussion

In summary, we presented the single-molecule control between two robust keto-enol tautomeric states in the single-molecule junctions, controlled by applying a moderate voltage at room temperature. The underlying mechanism of the presented device is based on a charge-injection-induced transition between two PESs, one with a preference for the (insulating) keto form and another with a preference for the (conducting) enol form, leading to significantly improved conductance difference between the tautomeric states. We used the MCBJ technique based on a microfabrication chip to demonstrate that the orientation of the single-molecule junction plays a critical role in the electrical control of the keto-enol tautomerization, indicating an electric-field contribution to the mechanism. The operation mechanism shows great potential for controlling both the kinetic barrier and thermodynamic driving force of a single-molecule reaction through the precise access of PES in both neutral and charged states. This leads to a single-molecule device with high-conductance ratios (up to a factor of 67), rapid response times (within 0.1 ms), and robust transformation behavior at room temperature. While we have demonstrated that the keto-enol equilibrium can also be modulated using a conventional electrochemical approach, the two-terminal junction system offers selective and precise manipulation of individual molecules, which has the potential for many future applications. As such, the design principle presented here relies on a common functional unit, which suggests the promising potential of utilizing a two-terminal junction system to explore the voltage control of single-molecule reactions on both neutral and charged PESs.

## Methods

### Conductance measurements

The single-molecule conductance was measured the STM-BJ technique with a home-built setup[43,44]. In brief, a gold substrate was fabricated by depositing Cr/Au (10/100 nm) onto a silicon wafer with 300 nm silicon dioxide. The gold substrate was cleaned with piranha solution before the experiment. A gold tip was made by flame cleaning to form a gold bead. We circularly drove the gold tip in and out of contact with the gold substrate and continuously recorded the current signal with a fixed bias voltage. During the breaking of the gold-gold contact, a molecule can bridge the in situ formed nanogap electrodes, leading to a conductance plateau. We used a sampling rate of 20 kHz to collect more than 2000 traces to construct the conductance histograms.

### State switching

The switching was performed by directly changing the bias voltage between 0.1 and 0.6 V. We collected approximately 1,000 traces in each cycle step to construct the 1D conductance histograms. The peak center was determined by Gaussian fitting. The error was defined as the standard deviation of the Gaussian fitting.

### I-V characterization

The I-V characterization was modified from a previous protocol[28,45]. During the break junction process with a 0.1 V bias applied, once we observed conductance plateaus formed between $10^{-4}$ and $10^{-5}$ $G_0$, we stopped the movement of the gold tip and scanned the bias between −1 and +1 V. Before and after the voltage scan, we fixed the bias to 0.2 V and checked whether the measured conductance was still located between $10^{-4}$ and $10^{-5}$ $G_0$. We then selected those I-V traces in which the conductance remained between $10^{-4}$ and $10^{-5}$ $G_0$ before and after the voltage scan. We use a 200 × 200 matrix to plot the 2D I-V histograms. The matrix is achieved by evenly spacing the current (on a logarithmic scale) and voltage (on a linear scale). Each I-V trace is analyzed in this matrix, which is added up with all the matrixes of the I-V traces to get the matrix of the 2D conductance histogram, presenting the data density in different colors according to the color scale. We used the Gaussian function to fit each column of the 2D conductance histogram to obtain the Gaussian peaks, leading to the fitting line.

### The probability of high- and low-conductance states

We used an auto-classification algorithm based on spectral clustering to determine the ratio of the traces showing different conductance plateau values[46]. When the bias is between 0.5 and 0.7 V, the conductance traces are clustered into two clusters, showing two clear molecular peaks located at approximately $10^{-4.5}$ and $10^{-2.8}$ $G_0$ in the corresponding 1D conductance histograms. The probabilities are determined by the number of traces in one of the clusters over the total number of traces. When the bias is between 0.1 and 0.4 V, the number of high-conductance traces is low. We cluster the conductance traces into 5 clusters and take one of the clusters showing a clear conductance peak around $10^{-3.0}$ $G_0$ as the high-conductance state. We also remove the clusters that do not show conductance peaks between $10^{-0.5}$ and $10^{-5.5}$ $G_0$. The residual clusters are determined to be the low-conductance state. The corresponding probabilities are determined by the number of high- or low-conductance traces over the total number of conductance traces.

### Synthesis

All the syntheses were performed with standard Schlenk techniques. The inert atmosphere is nitrogen. The used reagents and solvents were received from commercial sources and directly used without further purification. The NMR spectra were collected from Bruker AV-400 (400 MHz) or Bruker-500 (500 MHz) spectrometers. The chemical shift of $^1H$ and $^{13}C$ is relative to tetramethylsilane. The high-resolution mass spectra were collected by a Bruker En Apex Ultra 7.0 T Fourier Transform Mass Spectrometer. The theoretical molecular mass was calculated by Compass Isotope Pattern software supplied by Bruker Co. The synthesis and characterization details are shown in Supplementary Discussions.

### MCBJ experiment based on microfabrication chip

The MCBJ chip was obtained by microfabrication (see details[47] in Supplementary Fig. 29). To conduct the MCBJ experiment, we fixed the MCBJ chip with an underneath pushing pod and two counter supports. After adding the solution of molecule **1** (0.1 mM) to the center of the gold electrode on the MCBJ chip, we broke the gold-gold contact into an adjustable nanogap by moving the pushing pod driven by a motor. The single-molecule junction can be formed after forming the adjustable electrode nanogap. We kept connecting and breaking the gold electrodes until the value of the conductance stayed between $10^{-4}$ and $10^{-5}$ $G_0$. The movement of the pushing pod was stopped. Then, the I-V characterization was performed and analyzed using the same method as the STM-BJ experiment.

### DFT calculations

All geometry optimizations were performed at the B3LYP/def2-TZVP level with implicit inclusion of the apolar solvent through a PCM model in GAUSSIAN09[48]. Since the actual solvent used in the experiment, 1,2,4-trichlorobenzene ($\varepsilon = 2.25$), is not available as a predefined solvent in GAUSSIAN09, the very similar tetrachloroethene ($\varepsilon = 2.5$) was used instead. A small electric field of 0.05 V Å$^{-1}$—which corresponds approximately to the effective field experienced by the molecular bridge due to the threshold bias applied between the two Au-contacts —was included in the calculations with the help of the "Field = $M \pm N$" keyword (see Fig. 5a for the field direction; the GAUSSIAN convention was used here)[16,49]. Due to the low field strength and the low polarity of the solvent, one can expect the solvent screening to be negligible[50].

### Transport calculations

Transport calculations were performed using the non-equilibrium Green's function (NEGF) method combined with DFT as implemented in the Artaios code[51,52], a post-processing tool for GAUSSIAN09. In the first step, gold pyramids containing 4 Au-atoms each in an fcc-gold (111) arrangement were connected to the individual sulfur atoms of the optimized bridge geometry, after which the resulting system was re-optimized with the B3LYP functional (the basis set was set to Lanl2MB on the Au atoms and to def2-TZVP for all the other elements) until forces lower than 0.05 eV Å$^{-1}$ were obtained for at least three consecutive optimization steps. During this second optimization, the internal coordinates of the Au-pyramids were kept frozen to prevent their disintegration. Once relaxed geometries were obtained this way, the pyramidal fcc-contacts were expanded to 20 atoms on each side. For the resulting structures, single-point calculations were performed at the B3LYP/LanL2MB level of theory (a minimal basis set was chosen to avoid ghost transmission and excessive through-space coupling between the contacts), again using the GAUSSIAN09 software[52]. In a final step, the Hamiltonian and overlap matrices were extracted to carry out the NEGF calculation within the wide-band-limit (WBL) approximation using the post-processing tool Artaios. In the WBL approximation, a constant value of 0.036 eV$^{-1}$ for the local density of states (LDOS) of the electrode surface was used. This value was taken from the literature[53].

### Reporting summary

Further information on research design is available in the Nature Portfolio Reporting Summary linked to this article.

## Data availability

The data that support the findings of this study are available from the corresponding author upon request.

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

## Acknowledgements

This work was supported by the National Key R&D Program of China (No. 22250003 (W.H.), 21722305 (W.H.), 21933012 (J.L.) and 22173075 (J.L.)) and the Fundamental Research Funds for the Central Universities (No. 20720220020 (J.L.), 20720220072 (J.L.), 20720200068 (J.L.), 20720190002 (W.H.)), Financial Support for Outstanding Talents Training Fund in Shenzhen, and the Fundamental Research Funds for Xiamen University (No. 20720190002). S.S. is supported by the Israel Science Foundation (ISF 520/18). T.S. acknowledges the Research Foundation-Flanders (FWO) for a position as a postdoctoral research fellow (1203419N) and the French National Agency for Research (ANR) for a CPJ grant.

## Author contributions

W.H., S.S., H.X., and C.T. conceived the idea for the paper; C.T. synthesized and characterized the molecules under the supervision of H.X.; C.T., T.L., Y.Y., and T.G. conducted the conductance characterization under the supervision of H.W. and J.L.; T.S. and S.S. conducted the theoretical calculations. J.Z. and W.L. made the microfabrication MCBJ chip; C.T., T.S., J.L., L.L., J.S., S.S., H.X., and W.H. analyzed the data and wrote the paper with input from all the authors.

## Competing interests

The authors declare no competing interests.
