## [Peer review file · Nature Communications]

REVIEWER COMMENTS

Reviewer #1 (Remarks to the Author):

This is an interesting study that reports a single molecule conductance switch based on keto-enol tautomerisation reaction at the single molecule level. The study is complete, most of the experimental results are highly convincing, and appropriate control experiments were conducted. The theory is elegant and well explained. The text is very well-written, and I really enjoyed reading it.

I therefore recommend publication after the authors address the following:

A central point of the paper is the mechanism of the tautomerisation reaction involving charge injection, otherwise the barrier for separating the keto and enol forms is beyond reach. I think the authors should discuss the cyclic voltammetry of the SI (Figure S14) in the main text as it helps showing that the oxidation states are accessible. I also think the CVs should be explained in more details. For example, it appears that the CV in Fig S14 A shows three reduction peaks. These peaks should be explained. I also recommend adding the reference electrode in the CV's x-axis and plotting current density on the y-axis instead of current.

Text line 150 should be absorption instead of adsorption.

Figure S15 explanation is too brief in the SI.

It appears to me that the length of the plateaus in Figure 2C is less than that in Figure 2B. If that is true, does the magnitude of the bias affect the plateau length and the mechanical stability of the circuit? Are the counts in Figure 2d normalised such that one can see if there is a drop in counts with the applied bias?

I am wondering what will happen if the STMBJ experiments were carried out at -0.6 V? One would expect the keto state to prevail? I think inverting the sign of the bias is needed to strengthen the conclusions of the paper.

Reviewer #2 (Remarks to the Author):

Hong and co-workers demonstrate kinetic and thermodynamic control of keto-enol tautomerisation by external electric fields. Using a single-molecule junction set-up, they oxidise the parent system, reducing the tautomerisation barrier and shifting the equilibrium towards an otherwise disfavoured enol form. The latter affords continuous π -conjugation across the molecular bridge, leading to significantly higher conductance of the junction. Experimental observations are supported by theoretical analysis of the reaction energetics (at the DFT level) and transmissions spectra (at the NEGF-DFT level under the WBL approximation). The manuscript is well-written, and the supporting information is sufficiently detailed. My main concern is that this work is a rather niche example of manipulating reaction energetics with EEFs for a carefully chosen reaction. Practical utility of this approach at scale and appeal to the broad readership of Nature Communications are limited, therefore I recommend publication in a more specialised journal. My specific comments are as follows:

1. It is not rigorous from the quantum mechanics perspective to equate keto-enol tautomerisation in the parent and oxidised states. In this work, it is not the parent reaction that is manipulated, rather it is an entirely different equilibrium - in a system which lacks one electron compared to the parent one - that is being affected. In other words, the tautomerisation in the neutral state is not modified via subtle induction and electrostatic effects. Instead, the system is oxidised at high voltage within an SMJ set-up; the barrier is lower and the equilibrium is shifted towards enol in the resulting radical ion state. Arguably, the same could be achieved in a CV set-up instead.
2. SMJs are typically employed to introduce directionality, i.e. align the reactive system with respect to the field. Directionality is evoked in this study as well, however, what is shown experimentally is instead the order of oxidationreduction vs reductionoxidation.
3. How likely is this approach to become practical and scalable? How comparable is it to conventional synthetic methods of manipulating this particular reaction? The authors might argue that their goal is to demonstrate the concept rather than develop an industrially relevant approach. In this case, in my opinion, the novelty of this concept is limited.

Reviewer #3 (Remarks to the Author):

This manuscript by Tang et al. presents a neat method for controlling keto-enol tautomerism through the scanning tunneling microscope break junction (STM-BJ) technique. While the tested molecule strongly favors the keto form under normal conditions, one-electron oxidation led by a bias voltage encourages the enol form by switching to a potential energy surface (PES) of the cationic state(s). Many impressive experiments have been dedicated to confirming this general picture and ruling out alternative hypotheses; efforts pouring into this study are obvious and complete. Calculated energetic profiles along possible reaction coordinates and prior literature (notably ref. 36-38) are also consistent with the notion of redox-controlled keto-enol equilibrium. The potential application of single-molecule

rectifiers is also quite amazing. The main text is clearly written (especially for me who is not an expert in STM-BJ-related techniques), figures are well made, and the interpretations for each experimental result are convincing. I believe this manuscript is of fundamental importance and I am in favor of its publication. However, I have a few concerns that the authors should probably address before acceptance:

(1) The authors have made convincing arguments that the keto-enol equilibrium is perturbed by one-electron redox. However, the role of the electric field in this study to me seems secondary if not unimportant compared to the switch between different PESs. With the small dipole moment differences between the transition state and the starting state ($1.1 e \text{ \AA}$ and $0.7 e \text{ \AA}$ for the cationic and neutral state, respectively, inferred from computational results in Figs. 5 and S6), an intense electric field of 0.05 V/\AA seems to play a minor role (1.3 kcal/mol difference at most in activation free energy between field on and field off) in stabilizing/destabilizing the transition state compared to the much stronger effect from one-electron redox ($\sim 25 \text{ kcal/mol}$ difference). In fact, if the electric field effect were to be important, I would expect the two orientations described in Figure 4 to experience opposite electric fields, and 50% of the molecules would be anti-catalyzed rather than catalyzed. This would lead to at most 50% conversion to the enol form and be contradictory to the result from 0.7-V bias voltage in Figure 2f. Clearly electric fields alone cannot achieve the same effect if not accompanied by oxidation. Given extensive mention of oriented external electric fields (OEEFs) throughout the manuscript, I wonder if the authors can better justify the electric field effect, or the narrative can be confusing if not misleading. I also suggest the title be changed to a more unambiguous one, such as "Redox control of single-molecule keto-enol tautomerism" to reflect the dominant effect.

(2) While other pieces of evidence are persuasive, I am not quite convinced by controls in the glove box (to remove water) and isotope effects. While I believe the authors have done their best efforts to remove moisture to maybe sub ppm level, I am not sure if it is entirely feasible to remove any trace amount of water that is relevant on the single-molecule level. If the authors would like to really rule out any effect from catalytic water, I wonder if it is possible to instead humidify the environment without harming the STM-BJ conductance. The water-catalyzed process (Figure S7) could slightly better reconcile the huge barriers from Figure 5a with experimental observations. On the isotope effects, I believe the conductance distribution measured in Figure S1 reflects thermal equilibrium. No kinetic information is inferred (unless the conductance itself carries kinetic information as in some electrochemical methods), and proton tunneling, which should predominantly manifest as a kinetic isotope effect, cannot be ruled out. Also based on the extensive studies on proton-coupled electron transfer (PCET), it is certainly possible that the proton movement is coupled to the electron transfer to/from the electrode.

Minor issues:

(1) Page 1, line 17: the word "insurmountable" has been used several times and I find it a bit subjective. Is it possible to quote an energy for the isomerization barrier?

(2) Page 1, lines 27-28: "...to significantly modulate the electronic coupling without involving significant changes in the molecular frame..." I do not think I understand what "electronic coupling", which is a very specific term in Marcus theory, means in this context. Does this mean "electronic distribution"? The same confusion applies to "molecular frame", which can be a reference frame from the perspective of a molecule (as opposed to the "lab frame"). I believe the term stands for "nuclear configuration" or "nuclear coordinates". Consider better phrases as suggested.

- (3) Page 1, line 30: “molecular frames”. See previous comment.
- (4) Page 1, lines 33-38: “... with a lower isomerization barrier.... the isomerization barrier would be too high ...”. I am a bit confused by these two sentences. Aren’t they contradictory to each other?
- (5) Page 2, lines 45-51. OEEFs can be confusing in this work (see above). While I understand past works have been done with STM-BJ techniques (especially the Venkataraman lab and the authors’ lab) to verify the power of OEEFs, this work is probably not one of them. Also the notion of “external” in OEEFs does not seem to apply to the enzymatic processes, where the electric field is exerted via electrostatic distributions of charges from atoms rather than externally applied (through a capacitor and/or STM-BJ).
- (6) Page 2, line 48: “electrical control”. “Electrical” is a vague term as discussed above. Does it refer to “redox” or “electric field”? Are both aspects not well established, or is the single-molecule aspect not well-established?
- (7) Page 2, line 50: “... design the PESs...”. Consider “modulate” or “perturb”. I am not sure if a design process is really involved here.
- (8) Page 2, line 50: “... charge”. Should be “charged”.
- (9) Page 2, line 65: “... modulation of the applied voltage...”. I think it is useful to bring up STM-BJ at this point (and Figure 1 already explicitly shows it), as opposed to only bringing this technique up in line 98.
- (10) Page 3, lines 98 – 102. It helps to explain more regarding the interpretation of those plateaus. As I am not familiar with this technique, it took me a while to realize what they mean.
- (11) Page 5, line 150: “adsorption”. Should be “absorption”.
- (12) Page 6, line 172: “distance”. This term should be “energy difference” or “energy gap” instead.
- (13) Page 6, line 198: “respectively”. I am not sure if I understand what this term refers to. Do the voltages correspond to anything mentioned before?
- (14) Page 8, line 233: “ground-state”. This term might not be necessary as no photochemistry is involved.
- (15) Figure 5: If electric field effect is truly important, it helps to bring the reversed field configuration (Fig. S6) to the main text (or drawn together).

Response Letter to Reviewers' Comments

Revision Performed according to the Reviewers' Comments (revised text/words are highlighted in the MS and SI):

Reply to Reviewer #1

Comments: This is an interesting study that reports a single molecule conductance switch based on keto-enol tautomerisation reaction at the single molecule level. The study is complete, most of the experimental results are highly convincing, and appropriate control experiments were conducted. The theory is elegant and well explained. The text is very well-written, and I really enjoyed reading it.

Reply: We thank the reviewer for providing positive feedback and constructive suggestions on our work. In response to the reviewer's comments, we have revised the manuscript and included additional data and discussions.

Comment 1: I therefore recommend publication after the authors address the following:

A central point of the paper is the mechanism of the tautomerisation reaction involving charge injection, otherwise the barrier for separating the keto and enol forms is beyond reach. I think the authors should discuss the cyclic voltammetry of the SI (Figure S14) in the main text as it helps showing that the oxidation states are accessible. I also think the CVs should be explained in more details. For example, it appears that the CV in Fig S14 A shows three reduction peaks. These peaks should be explained. I also recommend adding the reference electrode in the CV's x-axis and plotting current density on the y-axis instead of current.

Reply 1-1: We thank the reviewer for the comment. According to the suggestion, we have revised the x- and y-axis of Figure S15A (Figure S14 in the previous version) by including the reference electrode and current density (as shown below).

Figure S15. The electrochemical STM-BJ experiments for compound 1. (A) The cyclic voltammetry is characterized in the propylene carbonate with 1.0 mM molecule 1 and 0.1 M tetrabutylammonium hexafluorophosphate as the electrolyte. (B-E) The 1D conductance histograms in 0, 0.3, 1.0, and 1.3 V EC gates, respectively. All the above conductance measurements are performed under ambient conditions with 0.1 V bias applied. (F) The 2D conductance histograms in 1.0 V EC gate.

We agree with the reviewer there are three reduction peaks, appearing at around 0.3, 0.7, and 1.2 V in the reductive scan, while there are two oxidation peaks, appearing at 1.0 and 1.3 V in the oxidative scan. The reductive peaks at 0.7 and 1.2 V should be the redox pairs corresponding to the oxidation peaks at 1.0 and 1.3 V. We think the remaining reduction peak at 0.3 V suggests that there is significant structural reorganization, i.e., keto-enol tautomerization, happening in the oxidative scan. As shown in Figure S16 (Figure S15 in the previous version), we used UV-Vis to characterize the oxidation species of 1 at 1.0 V, and found a redshift absorption, which is consistent with the methylated enol form **2-OMe**. This result suggests that the structure relaxation after oxidation can lead to a species with a smaller bandgap, which will have a different electrochemical response to the original state and lead to a new reduction peak.

Figure S16. The UV/Vis spectra for the electrochemical product. The UV/Vis spectra of **1** in 0 V and 1.0 V EC gate, with the molecule **2-OMe** as reference. All the UV/Vis spectra are characterized with 0.1 mM target molecules in a solution of propylene carbonate with 0.1 M tetrabutylammonium hexafluorophosphate added.

Following the reviewer's suggestion, we expanded the corresponding discussion in both the main text and SI:

In the main text, on page 5, line 58,

*"As a control experiment, we also performed the electrochemically gated²⁷ STM-BJ experiment on **1** in a four-electrode system (Supplementary Fig. 15), proving that the conductance state after one-electron oxidation of **1** is similar to the '**H**' state achieved at a higher bias (Fig. 2d). If there is a transformation from keto to enol form, we would expect a change of band gap since the enol form is a more conjugated structure. As characterized by the UV-Vis absorption (see Supplementary Fig. 16), we found that the one-electron oxidized species of **1** shows a red-shifted absorption similar to **2-OMe**, reinforcing our hypothesis that the high conductance state should be the one-electron oxidized enol form."*

In the SI, we added some more discussion as well,

*"We have performed an electrochemical (EC) STM-BJ experiment in a four-electrode system¹³ to oxidize the molecule in-situ and simultaneously measure the corresponding single-molecule conductance (cf. Figure S15). The ECSTM-BJ was performed with 1.0 mM molecule **1** and 0.1 M tetrabutylammonium hexafluorophosphate as the electrolyte. The working, reference, and counter electrodes are the gold tip, Ag/AgCl, and platinum wire. In the conductance characterization, the gold tip was coated with Apiezon wax³ to reduce background capacitive current and electrochemical currents."*

As shown in Figure S15A, the cyclic voltammetry characterization was started at -0.2 V with an oxidative scan by our ECSTM-BJ setup. In the oxidative scan, the cyclic voltammetry of molecule **I** shows two consecutive oxidation peaks at around 1.0 V and 1.3 V EC potentials (relative to Ag/Cl), which correspond to the one-electron and two-electron oxidization and suggest that oxidation states are accessible. We observed three reduction peaks at around 0.3, 0.7, and 1.2 V in the reductive scan. The reductive peaks at 0.7 and 1.2 V should be the redox pairs corresponding to oxidation peaks at 1.0 and 1.3 V. The new reductive peak that appeared in the reductive scan at 0.3 V suggests that there was a significant structural reorganization in the oxidative scan.

We use an EC gate and measure the single-molecule conductance simultaneously. All the conductance characterizations are performed with 0.1 V bias applied between the tip and substrate. As shown in Figure S15B, the 0 V EC gate measurement reveals a mono-conductance state around $10^{-4.5} G_0$, which is the low-conductance state (keto form) of molecule **I**. At the 0.3 V EC gate (Figure S15C), there was only the signal of the low-conductance state. When applying a 1.0 V EC gate, we observe two distinctive conductance peaks, centering around $10^{-2.0} G_0$ and $10^{-2.8} G_0$ (Figure S15D). Its 2D conductance histogram indicates that the conductance plateaus of $10^{-2.8} G_0$ are about two times longer than the plateaus of $10^{-2.0} G_0$ (shown in Figure S15F), while the two types of plateaus occur consecutively. The conductance plateaus of $10^{-2.8} G_0$ are very similar to the high-conductance state (enol form) of the two-electrode measurement at 0.6 V (Figure 2d). We also observe that upon two-electron oxidation, a low-conductance state is reached again (shown in Figure S15E). This state most likely corresponds to a deprotonated keto form."

Comment 2: Text line 150 should be absorption instead of adsorption.

Reply 1-2: We thank the reviewer for the suggestion. We have revised the context as suggested.

Comment 3: Figure S15 explanation is too brief in the SI.

Reply 1-3: We thank the reviewer for the suggestion. We expanded the corresponding discussion of Figure S16 (Figure S15 in the previous version) in SI as suggested,

"To understand the structure in oxidation, we also measured the UV/Vis spectra of the oxidized species. We expect that the enol form will have a more extended pi-system than that of the corresponding keto form. Therefore, the enol form will have a smaller bandgap than that of the keto form. As expected, the methylated enol form **I-OMe** (with the extended pi-system) exhibits a redshift absorption (50 nm) with respect to molecule **I** (grey curve). More importantly, after one one-electron oxidation in molecule **I**, we observed a similar redshift absorption (red curve), which

is consistent with the absorption of **1-OMe**. The UV/Vis spectra further support that the enol form was generated from the keto form upon one-electron oxidation and provide additional proof that the high-conductance state indeed corresponds to an enol structure."

Comment 4: It appears to me that the length of the plateaus in Figure 2C is less than that in Figure 2B. If that is true, does the magnitude of the bias affect the plateau length and the mechanical stability of the circuit? Are the counts in Figure 2d normalised such that one can see if there is a drop in counts with the applied bias?

Reply 1-4: We thank the reviewer for the comment. After carefully analyzing the conductance plateaus, we found that the length of the plateau in Figure 2C is slightly longer than that of Figure 2B (0.47 nm vs. 0.42 nm), though there is a bigger spread in the measured lengths due to the simultaneous presence of both the high- and low-conducting state in the plot (*vide infra*), resulting in a less clearly discernable plateau.

This point can be better understood by splitting the two-dimensional histograms at 0.6 V bias in Figure 2C according to whether they belong to the high-conductance or low-conductance states. As shown in Figure S27, upon separation, we can observe that the high-conductance states have more clearly longer conductance plateaus (Figure S27B) than the low-conductance states (Figure S27D). Moreover, the conductance plateaus of **1** at 0.6 V bias (Figure S27B) are almost identical to that of **2-OMe**, both in length and conductance values (Figure S4).

Figure S27. The experimental data at 0.6 V bias. The 1D conductance histograms of the high-conductance state (A) and low-conductance state (C). The 2D conductance histograms of the high-conductance state (B) and low-conductance state (D) with the insets showing the stretching distance ranging from $10^{-6.0}$ to $10^{-0.3} G_0$. Traces are assigned to the high-conductance state if they

retain a conductance ranging from $10^{-3.2}$ to $10^{-1.1} G_0$ for 0.2 nm longer than average. The remaining traces are attributed to the low-conductance state. 5.1% of the traces were excluded due to excessive noise.

Figure S4. 1D and 2D conductance histograms of 2-OMe. All the STM-BJ experiments were performed on a 0.1 mM solution of 2-OMe with the bias changed from 0.1 to 0.6 V.

We also tried to normalize the counts in Figure 2d by the number of conductance traces but did not find a clear dependence on the bias. Therefore, we think that changing the bias magnitude did not significantly affect the mechanical stability of the junction within the bias range from 0.1 to 0.6 V. To make this point more clear, we have added the following discussion at line 152, "*Meanwhile, the conductance plateau of 2-OMe (Supplementary Fig. 4) is slightly longer than the 'L' state and is similar to the 'H' state (Supplementary Fig. 27), suggesting that the 'H' state has a similar junction geometry to 2-OMe within the bias ranging from 0.1 to 0.6V.*".

Comment 5: I am wondering what will happen if the STMBJ experiments were carried out at -0.6 V? One would expect the keto state to prevail? I think inverting the sign of the bias is needed to strengthen the conclusions of the paper.

Reply 1-5: We thank the reviewer for the suggestion. As suggested, we performed the STMBJ experiment on compound **1** with a negative bias voltage applied, the result of which is added in Fig. S6. As shown in Fig. S6A, the 1D conductance histograms show a similar trend to the experiment performed with the positive bias (Fig. 2d). Namely, the higher magnitude of applied bias will lead to a higher proportion of the high-conductance state (Fig. 2f and Fig. S6B). This result suggests that the tip and substrate will shift evenly in opposite directions relative to the Fermi level (E_F) when a certain magnitude of bias (ΔE) is applied (Fig. R1), which is the circumstance in many two-

terminal junction systems. Although molecular component **1** has asymmetric factors, one cannot expect this to show up in this experimental setup since the molecular orientation is still random across a population of molecular junctions formed. Therefore, the experiment with a negative bias will inevitably show the same response as the experiment with a positive bias applied (when considering a single junction with a fixed orientation, however, we do observe a directional effect, cf. the discussion of the microfabrication chip experiments below). To make this point clear, we revised the discussions on Page 6, line 180, "*It is worth noting that the current jumps happened in both positive and negative bias. This observation is consistent with the finding that the 'H' states become more dominant as the magnitude of the bias increases, regardless of whether the bias is positive (Fig. 2) or negative (Supplementary Fig. 6). As such, we used the absolute bias value to construct the 2D I-V histogram for clarity (Fig. 3a)*".

Figure S6. Control experiments in negative bias. (A) ID conductance histograms of **1** were obtained with different biases. The STM-BJ experiments are performed on the 0.1 mM solution of **1** in TCB under ambient conditions. (B) The distribution probabilities of states 'L' and 'H' are plotted against different biases.

Figure R1. The representation diagrams illustrate the shifting of chemical potential in the tip (T) and the substrate (S) in opposite directions when a positive bias (ΔE) and negative bias ($-\Delta E$) are applied, respectively. This shift occurs in a symmetrical manner relative to the Fermi level (E_F).

Reply to Reviewer #2

Comments: Hong and co-workers demonstrate kinetic and thermodynamic control of keto-enol tautomerisation by external electric fields. Using a single-molecule junction setup, they oxidise the parent system, reducing the tautomerisation barrier and shifting the equilibrium towards an otherwise disfavoured enol form. The latter affords continuous π -conjugation across the molecular bridge, leading to significantly higher conductance of the junction. Experimental observations are supported by theoretical analysis of the reaction energetics (at the DFT level) and transmissions spectra (at the NEGF-DFT level under the WBL approximation). The manuscript is well-written, and the supporting information is sufficiently detailed. My main concern is that this work is a rather niche example of manipulating reaction energetics with EEFs for a carefully chosen reaction. Practical utility of this approach at scale and appeal to the broad readership of Nature Communications are limited, therefore I recommend publication in a more specialised journal.

Reply: We thank the reviewer for providing valuable suggestions for our research and recognizing the effort we have put into this work. We are sorry that we did not effectively convey the key points of our work to the reviewers to fully recognize its broad interest. We have taken great care to revise this manuscript according to the reviewer's suggestions. Specifically, we have revised corresponding discussions to ensure that the key points of our research are more effectively communicated. The revision details are shown in the following responses to the reviewer's comments.

Comment 1: My specific comments are as follows: It is not rigorous from the quantum mechanics perspective to equate keto-enol tautomerisation in the parent and oxidised states. In this work, it is not the parent reaction that is manipulated, rather it is an entirely different equilibrium - in a system which lacks one electron compared to the parent one - that is being affected. In other words, the tautomerisation in the neutral state is not modified via subtle induction and electrostatic effects. Instead, the system is oxidised at high voltage within an SMJ setup; the barrier is lower and the equilibrium is shifted towards enol in the resulting radical ion state. Arguably, the same could be achieved in a CV set-up instead.

Reply 2-1: We agree with the reviewer's comment that the thermodynamic equilibrium of the keto-enol tautomerization is not manipulated in the parent state. Instead, the equilibrium is modulated in the one-electron oxidized state. This is also a key point of this work, as discussed in the abstract,

"This work highlights the concept of single-molecule control of chemical reactions on more than one potential energy surface."

In order to eliminate any confusion between the electrical modulation of keto-enol equilibrium and the concept of tautomerism, we have modified the title from "*Electrical control of single-molecule keto-enol tautomerism*" to "*Voltage-driven control of single-molecule keto-enol equilibrium in a two-terminal junction system*". This change was made based on our appreciation of the reviewer's comment, and we think the term equilibrium can more accurately describe the observed phenomenon.

We also agree with the reviewer's concern that such equilibrium manipulation can also be achieved in a regular CV setup. However, achieving this manipulation in a two-terminal junction system instead is of significant importance for a wide variety of reasons:

1. Although many single-molecule scale reactions have been demonstrated in the CV setup, very few examples exist of such control at the single-molecule level in a two-terminal junction system. Therefore, this research will provide an important foundation for the discovery of more reactions that can be controlled in a two-terminal junction in further studies.
2. The two-terminal junction system can selectively and precisely manipulate the reaction at the single-molecule scale. In a CV setup, all the molecular wires inside the electrolyte need to be oxidized or reduced, but only one molecule will be connected between the electrodes in a junction system. As such, when considering an electronic application, the CV setup inherently loses energy efficiency because most of the molecules will not be used for constructing the molecule device.
3. The two-terminal junction system has its own advantage in achieving manipulation in a high spatial resolution. For example, when considering pattern generation, the two-electrode system can be directly fitted with dip-pen nanolithography to achieve the desired pattern in a high spatial resolution on self-assembled monolayers, which is not feasible in the CV setup.
4. The two-terminal junction system has a promising potential to achieve a high level of integration in small device sizes. In the CV setup, each device needs four electrodes, namely two working electrodes, one reference electrode, and one counter electrode. Meanwhile, using electrolytes is necessary. It is a great challenge to shrink the size of individual electrochemical devices, considering the device complexity, which is also challenging when considering device integration. In contrast, the two-terminal junction

system is in a much-simplified structure, which will benefit the integration of molecule devices.

To distinguish the importance of our two-terminal junction system to the conventional electrochemical system, we added a brief discussion in the conclusion part on page 11, line 325, *"While we have demonstrated that the keto-enol equilibrium can also be modulated using a conventional electrochemical approach, the two-terminal junction system offers selective and precise manipulation of individual molecules, which has the potential for many future applications."*

Comment 2: SMJs are typically employed to introduce directionality, i.e. align the reactive system with respect to the field. Directionality is evoked in this study as well, however, what is shown experimentally is instead the order of oxidationreduction vs reductionoxidation.

Reply 2-2: We thank the reviewer's comment. We agree with the reviewer that the redox effect is very important, but at the same time, the relative orientation between the molecular component and the external field is also not negligible. We have observed a polarized electronic behavior that cannot be fully explained by conventional electrochemical processes alone:

Based on the MCBJ experiment conducted on a microfabrication chip, as illustrated in Figure 4a, a single-molecule junction can be formed, and its lifetime can last for more than 10 minutes while the orientation of the molecule remains fixed. As demonstrated in Fig. 4b, the transformation behavior is observed only in the negative bias range, while the positive bias range exhibits a linear response. However, in another molecular device with a different molecular orientation, the transformation behavior occurs only in the positive bias range (Fig. 4e). If we consider only conventional electrochemical processes, both positive and negative biases should exhibit the exact same transformation behavior.

Figure 4 | Constructing single-molecule devices. *a*, Schematic of the MCBJ chip. The SEM image of the nanogap electrodes is shown with a 1 μm scale bar. *b*, 2D I-V histogram constructed from 1445 (out of 2302) traces exhibiting current jumps. The representative I-V curve is shown in the yellow line, which is replotted in the inset figure with the current value on a linear scale. *c*, Based on the representative I-V curve, the TVS is shown correspondingly for both the negative and positive bias. The absolute values of transition voltage are shown correspondingly. *d*, Part of a conductance trace shows the change of conductance during the switching of bias between 0.1 and 0.6 V. *e*, The conductance trace shows the robust switching behavior. The data within the dashed frame is zoomed in on panel 4d.

Such polarized behavior is associated with the oriented field effect. As shown in the revised Fig. 5a, we calculate the PES with the electric field applied in opposite directions. We find that a certain electric field direction can further reduce the reaction barrier, while the opposite field direction actually increases the barrier to a non-negligible extent (the total spread amounts to almost 3 kcal/mol). Moreover, the one-electron oxidative species are ionic compounds, the energy of which typically exhibits a stronger dependence on the electric field than the neutral species, and correspondingly, they are associated with the more significant barrier change with different field directions in the charged PES.

Figure 5 | *Calculated reaction profile and transmission coefficient. a, Reaction profile (in kcal mol⁻¹) associated with the tautomerization reaction on the uncharged PES (black), and on the PES of the radical, cationic species obtained after charge injection (red), calculated at B3LYP/def2-TZVP level of theory. An electric field corresponding to the field exerted by the approximate threshold voltage in the junction has been included in different directions (shown in orange and blue arrows). The PES in the different directions of the electric field is linked by orange (-0.05 V \AA^{-1}) and blue ($+0.05 \text{ V \AA}^{-1}$) dash lines.*

Therefore, the redox order and the relative orientation between the molecular component and the external field are two important factors for determining the barriers and thermodynamic equilibrium of the keto-enol tautomerization in the neutral and charged PES.

Comment 3: How likely is this approach to become practical and scalable? How comparable is it to conventional synthetic methods of manipulating this particular reaction? The authors might argue that their goal is to demonstrate the concept rather than develop an industrially relevant approach. In this case, in my opinion, the novelty of this concept is limited.

Reply 2-3: We thank the reviewer for this critical comment. In reply to the reviewer's concern, we expanded the discussions and included corresponding revisions to better illustrate the novelty of this work:

1. The reviewer commented, "How likely is this approach to become practical and scalable?". As we discussed above, controlling the chemical reaction in the two-terminal junction system has its own advantage for device integration, owing to the simple junction structure, which significantly reduces the device complexity for practical and scalable applications. Currently, there are two challenges for the practical and scalable application of single-molecule devices: the construction of highly dense electrode pairs with both high quality and uniformity; understanding the electronic performance of the molecular component to achieve good performance. In this work, we are focusing on the second challenge and achieving important progress. Namely, we have found a novel manner to modulate electronic behavior by controlling single-molecule reactions in a two-terminal junction system. This idea will stimulate more efforts in this direction. On the other hand, our device shows good performance, including a large on-off ratio, very fast switching speed within 0.1 ms, and robust behavior in ambient conditions. Meanwhile, the involved chemical reaction can generate rectification performance, which also provides a new strategy for designing molecular diodes. Therefore, our work demonstrates the promising idea of coupling a chemical reaction in a two-terminal junction system for designing unique molecule devices. To clarify this point, we added discussion in the conclusion part on page 11, line 325, "*While we have demonstrated that the keto-enol equilibrium can also be modulated using a conventional electrochemical approach, the two-terminal junction system enables selective and precise manipulation of individual molecules, which has the potential for many future applications.*", and one page 10, line 321, "*The operation mechanism shows great potential for controlling both the kinetic barrier and thermodynamic driving force of a single-molecule reaction through the precise access of PES in both neutral and charged states. This leads to a single-molecule device with a high conductance ratio (up to a factor 67), rapid response times (within 0.1 ms), and robust transformation behavior at room temperature.*". To emphasize the importance of the two-terminal junction system, we revised the title to "*Voltage-driven control of single-molecule keto-enol equilibrium in a two-terminal junction system*".
2. The reviewer commented, "How comparable is it to conventional synthetic methods of manipulating this particular reaction?". Firstly, we would like to underscore that this reaction is not a niche one. Keto-enol tautomerization is one of the most common chemical reactions (Antonov, L. *Tautomerism: methods and theories*. (Wiley-VCH, 2014). Based on the *Reaxys* database, at least 7 million reported compounds have the

keto-enol functional unit, which suggests the universality of utilizing the developed method to control chemical reactions. Secondly, if we target molecular synthesis, our method does not have the advantage for large-scale synthetic applications compared to conventional synthetic methods. Instead, the control strategy employed in a two-terminal junction system exhibits high spatial resolution and selectivity, enabling precise manipulation of single-molecule reaction, which will have advantages for nanoscale applications that cannot be achieved by conventional synthetic methods. Thirdly, our design principle has demonstrated the idea of controlling chemical reactions in both neutral and charged PES, which suggests our method is not only limited to the keto-enol tautomerization system. Instead, there are more abundant systems with redox activity that can be explored in this two-terminal junction system. To better clarify this point, we revised the conclusion part on page 11, line 327, "*As such, the design principle presented here relies on a common functional unit, which suggests the promising potential of utilizing a two-terminal junction system to explore the voltage control of single-molecule reactions on both neutral and charged PESs.*".

In brief, our research introduced a novel approach for regulating chemical reactions in a two-terminal junction system, exhibiting the potential of controlling reactions with single-molecule selectivity and high spatial resolution. We demonstrated the design principle using a common functional unit, and our findings suggest that it holds promising potential for being extended to a wider range of systems with redox activity. Therefore, we believe the concept we developed will attract broad interest.

Reply to Reviewer #3

Comments: This manuscript by Tang et al. presents a neat method for controlling keto-enol tautomerism through the scanning tunneling microscope break junction (STM-BJ) technique. While the tested molecule strongly favors the keto form under normal conditions, one-electron oxidation led by a bias voltage encourages the enol form by switching to a potential energy surface (PES) of the cationic state(s). Many impressive experiments have been dedicated to confirming this general picture and ruling out alternative hypotheses; efforts pouring into this study are obvious and complete. Calculated energetic profiles along possible reaction coordinates and prior literature (notably ref. 36-38) are also consistent with the notion of redox-controlled keto-enol equilibrium. The potential application of single-molecule rectifiers is also quite amazing. The main text is clearly

written (especially for me who is not an expert in STM-BJ-related techniques), figures are well made, and the interpretations for each experimental result are convincing. I believe this manuscript is of fundamental importance and I am in favor of its publication. However, I have a few concerns that the authors should probably address before acceptance:

Reply: We thank the reviewer for positively referring to our research and providing valuable suggestions. Following the reviewer's suggestions, we have carefully revised the manuscript and included additional data and discussions.

Comment 1: The authors have made convincing arguments that the keto-enol equilibrium is perturbed by one-electron redox. However, the role of the electric field in this study to me seems secondary if not unimportant compared to the switch between different PESs. With the small dipole moment differences between the transition state and the starting state ($1.1 e \text{ \AA}$ and $0.7 e \text{ \AA}$ for the cationic and neutral state, respectively, inferred from computational results in Figs. 5 and S6), an intense electric field of 0.05 V/\AA seems to play a minor role (1.3 kcal/mol difference at most in activation free energy between field on and field off) in stabilizing/destabilizing the transition state compared to the much stronger effect from one-electron redox ($\sim 25 \text{ kcal/mol}$ difference). In fact, if the electric field effect were to be important, I would expect the two orientations described in Figure 4 to experience opposite electric fields, and 50% of the molecules would be anti-catalyzed rather than catalyzed. This would lead to at most 50% conversion to the enol form and be contradictory to the result from 0.7-V bias voltage in Figure 2f. Clearly electric fields alone cannot achieve the same effect if not accompanied by oxidation. Given extensive mention of oriented external electric fields (OEEFs) throughout the manuscript, I wonder if the authors can better justify the electric field effect, or the narrative can be confusing if not misleading. I also suggest the title be changed to a more unambiguous one, such as "Redox control of single-molecule keto-enol tautomerism" to reflect the dominant effect.

Reply 3-1: We thank the reviewer for the comprehensive mechanism summary and for providing valuable suggestions. We agree with the reviewer that the "redox" effect is the most significant contributor to the observed phenomenon, and that the electric field plays only a secondary role (though it is responsible for the unique polarized behavior found in the microfabrication chip setup). Based on the suggestions made, we revised the title from "*Electrical control of single-molecule keto-enol tautomerism*" to "*Voltage-driven control of single-molecule keto-enol equilibrium in a two-terminal junction system*". We also followed the reviewer's suggestion to narrow the discussion of OEEFs to avoid confusion. We now only mention the OEEF effect as an example of a relevant

technique to modulate PESs in the introduction, on page 2, line 51, "*Additionally, oriented external electric fields (OEEFs)¹² have been used to reduce reaction barriers to accelerate chemical reactions, both in solution¹²⁻¹⁸ and at the single-molecule scale^{13,19}*", and invoke it to explain the results of the microfabrication chip experiments, on page 3, line 85, "*In the latter experimental setup, the alignment between the dipole moment of the molecular component and the electric field is observed to be critical for the electrically controlled tautomerization, i.e., the oriented electric field, resulting from the applied bias, acts as a non-negligible facilitator/inhibitor of the barrier crossing. This proposed control mechanism is further elucidated through theoretical calculations.*"

Comment 2: While other pieces of evidence are persuasive, I am not quite convinced by controls in the glove box (to remove water) and isotope effects. While I believe the authors have done their best efforts to remove moisture to maybe sub ppm level, I am not sure if it is entirely feasible to remove any trace amount of water that is relevant on the single-molecule level. If the authors would like to really rule out any effect from catalytic water, I wonder if it is possible to instead humidify the environment without harming the STM-BJ conductance. The water-catalyzed process (Figure S7) could slightly better reconcile the huge barriers from Figure 5a with experimental observations. On the isotope effects, I believe the conductance distribution measured in Figure S1 reflects thermal equilibrium. No kinetic information is inferred (unless the conductance itself carries kinetic information as in some electrochemical methods), and proton tunneling, which should predominantly manifest as a kinetic isotope effect, cannot be ruled out. Also based on the extensive studies on proton-coupled electron transfer (PCET), it is certainly possible that the proton movement is coupled to the electron transfer to/from the electrode.

Reply 3-2: We thank the reviewer for the constructive suggestion. Indeed, humidifying the environment in the STM-BJ experiment will provide a valuable clue to understanding the mechanism. Actually, we had made efforts by humidifying the environment to do the control STM-BJ experiments in our previous version (shown in Figure S2). We performed two control experiments in a solvent mixture of THF/TCB and in water-saturated TCB. THF is miscible with water. The THF we used has an initial moisture content of ~2%. Therefore, the solvent mixture of THF/TCB has a higher moisture content than TCB on its own. The water-saturated TCB was prepared by vigorously mixing with the equivalent volume of water, and stood for 24h. Then the TCB layer was taken out by pipette to perform the STM-BJ experiment. As shown in Figs. S2C and S2D, we observed a similar trend that the 'H' state becomes more and more dominant when a higher magnitude of the bias is applied in different humid solvents.

We agree with the reviewer's concern that the glove box control cannot remove any trace amount of water. Nevertheless, the control experiments in a humidified environment did not significantly change the STM-BJ experiments, suggesting that water may not significantly accelerate or inhibit the tautomerization process. This is also consistent with the theoretical calculations (Fig. S8). We found that water can decrease the tautomerization barrier. However, the introduced water did not change the reaction equilibrium, in which the keto form still dominates the reaction equilibrium. Therefore, only considering the water-introduced mechanism cannot explain our experimental results.

Figure S2. 1D conductance histograms of the water control experiments. (A) The experiment was performed in a glovebox and a dry TCB solvent. (B-E) The following STM-BJ experiments were performed in the corresponding solvents under the conditions shown above the 1D conductance histograms. The control experiments B-E, were performed under ambient conditions. Since the emergence of the 'H' peak in the conductance histograms does not appear to be affected by the presence/absence of water, the participation of this compound in the tautomerization mechanism can be ruled out. TMB is the abbreviation for sym-trimethylbenzene. All STM-BJ experiments were performed on a 0.1 mM solution of **I**.

We feel sorry that we did not expand the discussion enough to convey this result of control experiments effectively to the reader. To make this point clear, we expanded the discussion on Page 8, line 255, "*We performed control experiments in the glovebox (Supplementary Fig. 2a), and we also humidified the solvent environments (Supplementary Fig. 2b-e). Neither of these modifications to the reaction conditions changed the experimental results significantly, suggesting that H₂O does not significantly accelerate or inhibit the tautomerization process in the STM-BJ experiments.*".

We agree with the reviewer that a slight isotope effect is reflected in Fig. S1, in which **1-d₂** has about 4% lower distribution of the "**H**" state than that of **1** at 0.6 V. This is however, a very small effect, which allows us to conclude that the isotope effect between **1** and **1-d₂** is weak and thus we can rule out a proton tunneling mechanism (the isotope effect would need to be much bigger for that). To make this point clear, we revised the discussion on Page 9, line 259, "*The isotope effect is also proven experimentally to be weak (Supplementary Fig. 1), which rules out a proton tunneling mechanism.*". We added this sentence in the figure caption of Fig. S1, "*The '**H**' state of **1-d₂** shows a 4% lower distribution than that of **1** at 0.6 V.*".

Figure S8. Potential energy profiles associated with the rejected alternative H₂O-assisted reaction mechanism for the uncharged species (**1**) in the cases of $F_z = -0.05$ V/Å (blue) and $F_z = +0.05$ V/Å (red). Energies are denoted in kcal/mol.

Concerning the possibility of a PCET mechanism raised by the reviewer, DFT calculations do not enable a distinction between a PCET and a concerted/radical hydrogen transfer reaction. As

described at length by one of the authors of this manuscript on multiple occasions (e.g., <https://pubs.acs.org/doi/pdf/10.1021/ja408073m>), the reaction coordinate for both reaction modes generally coincide, so that transition states usually have some characteristics of both. Based on our Valence Bond analysis in Section S3 in the Supporting Information, we can estimate that the resonance mixing in the transition state region on the charged PES amounts to around 40 kcal/mol, which could be an indication that there is a non-negligible PCET admixture in the located TS. Nevertheless, it is clear that even in the highly unlikely case that the mechanism would be mainly PCET, there is still the computed barrier to cross.

Comment 3: Minor issues:

Page 1, line 17: the word "insurmountable" has been used several times and I find it a bit subjective. Is it possible to quote an energy for the isomerization barrier?

Reply 3-3: We thank the reviewer for the suggestion. We revised the sentence on Page 1, line 20, to be, "*...while a high isomerization barrier limits the transformation to the enol form...*", and revised the sentence on Page 2, line 40 to quote the energy, "*Some tautomerizations, however, have non-degenerate tautomeric states with the thermodynamic equilibrium strongly on one side, and the isomerization barrier to reach the other state is insurmountable (e.g., activation energies higher than 50 kcal mol⁻¹).*".

Comment 4: Page 1, lines 27-28: "...to significantly modulate the electronic coupling without involving significant changes in the molecular frame..." I do not think I understand what "electronic coupling", which is a very specific term in Marcus theory, means in this context. Does this mean "electronic distribution"? The same confusion applies to "molecular frame", which can be a reference frame from the perspective of a molecule (as opposed to the "lab frame"). I believe the term stands for "nuclear configuration" or "nuclear coordinates". Consider better phrases as suggested.

Reply 3-4: We thank the reviewer for the suggestion. We initially used the term "molecular frame" to represent molecular conformation, and use "electronic coupling between two electrodes" to represent the change of molecular conformation. To make this point clear, we revised the sentence on Page 1, lines 32-33 to, "*...Toward this goal, an ideal single-molecule reaction is expected to significantly modulate the electronic **properties** without involving significant changes in the molecular **conformation**...*".

Comment 5: Page 1, line 30: "molecular frames". See previous comment.

Reply 3-5: We thank the reviewer for the suggestion. We changed "molecular frames" to "*molecular conformations*".

Comment 6: Page 1, lines 33-38: "... with a lower isomerization barrier.... the isomerization barrier would be too high ...". I am a bit confused by these two sentences. Aren't they contradictory to each other?

Reply 3-6: We thank the reviewer for the suggestion. We revised the sentence to resolve the potential confusion as follows one page 2, line 40, "*Some tautomerizations, however, have non-degenerate tautomeric states with the thermodynamic equilibrium strongly on one side, and the isomerization barrier to reach the other state is insurmountable (e.g., activation energies higher than 50 kcal mol⁻¹). While the reaction equilibrium and high barrier can support a robust tautomeric state, they also limit the molecule's ability to switch.*".

Comment 7: Page 2, lines 45-51. OEEFs can be confusing in this work (see above). While I understand past works have been done with STM-BJ techniques (especially the Venkataraman lab and the authors' lab) to verify the power of OEEFs, this work is probably not one of them. Also the notion of "external" in OEEFs does not seem to apply to the enzymatic processes, where the electric field is exerted via electrostatic distributions of charges from atoms rather than externally applied (through a capacitor and/or STM-BJ).

Reply 3-7: We thank the reviewer for the suggestion. We agree with the reviewer that OEEF is not an optimal term to describe the electrostatic effects observed in enzymatic processes. To make this point clear, we revised the sentence on Page 2, line 53, "*Electric fields also play an important catalyzing role in many enzymatic processes^{18,20}.*".

Comment 8: Page 2, line 48: "electrical control". "Electrical" is a vague term as discussed above. Does it refer to "redox" or "electric field"? Are both aspects not well established, or is the single-molecule aspect not well-established?

Reply 3-8: We thank the reviewer for the suggestion. In the previous version, we used the "electrical control" to represent the combination of "redox control" and "electric field modulation". The combination of both aspects in a two-terminal junction system is not well established yet. We agree

with the reviewer that this the terminology used is not clear, and hence we have revised the title from "*Electrical control of single-molecule keto-enol tautomerism*" to "*Voltage-driven control of single-molecule keto-enol equilibrium in a two-terminal junction system*". Furthermore, we emphasize throughout the manuscript that the redox process plays a significantly more important role in this work than the electric field effect (though the latter is responsible for the rectification performance of the two-terminal junction system).

Comment 9: Page 2, line 50: "... design the PESs...". Consider "modulate" or "perturb". I am not sure if a design process is really involved here.

Reply 3-9: We thank the reviewer for the suggestion. The revised sentence is as follows, "... *we were able in this work to access the PESs in both the neutral and charged states, providing an opportunity to explore the potential of thermodynamic and kinetic control in single-molecule reactions...*".

Comment 10: Page 2, line 50: "... charge". Should be "charged".

Reply 3-10: We thank the reviewer for the suggestion. We have revised the context as suggested.

Comment 11: Page 2, line 65: "... modulation of the applied voltage...". I think it is useful to bring up STM-BJ at this point (and Figure 1 already explicitly shows it), as opposed to only bringing this technique up in line 98.

Reply 3-11: We thank the reviewer for the suggestion. We have moved the term STM-BJ up to Page 2, line 65, as suggested.

Comment 12: Page 3, lines 98 – 102. It helps to explain more regarding the interpretation of those plateaus. As I am not familiar with this technique, it took me a while to realize what they mean.

Reply 3-12: We thank the reviewer for the comment. We feel sorry that we neglected to expand the discussions related to the interpretation of those plateaus in the previous version. To improve this point, we added further discussions on Page 4, line 118, "*During a typical STM-BJ experiment, we continuously measure the conductance value while changing the distance between the gold tip and substrate. When a molecule bridges the gap between the two electrodes, the conductance change*".

in response to changes in the distance becomes less pronounced, resulting in conductance plateaus in the conductance traces (conductance vs. stretching distance)."

Comment 13: Page 5, line 150: "adsorption". Should be "absorption".

Reply 3-13: We thank the reviewer for the suggestion. We have revised the context as suggested.

Comment 14: Page 6, line 172: "distance". This term should be "energy difference" or "energy gap" instead.

Reply 3-14: We thank the reviewer for the suggestion. The revised sentence is as follows, "*An optimal agreement with the experimental data is obtained for an approximate energy difference between the Fermi level and the HOMO channel of 0.65–0.75 eV (cf. Fig. 3c).*".

Comment 15: Page 6, line 198: "respectively". I am not sure if I understand what this term refers to. Do the voltages correspond to anything mentioned before?

Reply 3-15: We thank the reviewer for the suggestion. We want to refer to the term transition voltage. Because of the polarized electronic behavior, the transition voltages obtained for the negative and positive bias regions are different. To make this point more precise, we have revised the sentence, "*...Transition voltage spectroscopy (TVS)³⁰ was utilized to determine the transition voltages in the negative and positive bias regions, which were found to be at 0.38 and 0.81 V, respectively (Fig. 4c).*".

Comment 16: Page 8, line 233: "ground-state". This term might not be necessary as no photochemistry is involved.

Reply 3-16: We thank the reviewer for the suggestion. We have removed the term related to "ground-state".

Comment 17: Figure 5: If electric field effect is truly important, it helps to bring the reversed field configuration (Fig. S6) to the main text (or drawn together).

Reply 3-17: We thank the reviewer for the suggestion. We have revised Figure 5a to combine the PES in both the forward and reverse fields.

Figure 5 | *Calculated reaction profile and transmission coefficient. a, Reaction profile (in kcal mol⁻¹) associated with the tautomerization reaction on the uncharged PES (black), and on the PES of the radical, cationic species obtained after charge injection (red), calculated at B3LYP/def2-TZVP level of theory. An electric field corresponding to the field exerted by the approximate threshold voltage in the junction has been included in different directions (shown in orange and blue arrows). The PES in the different directions of the electric field is linked by orange (-0.05 V \AA^{-1}) and blue ($+0.05 \text{ V \AA}^{-1}$) dashed lines.*

REVIEWERS' COMMENTS

Reviewer #1 (Remarks to the Author):

The authors have revised the manuscript adequately and have addressed my concerns. I recommend publication.

Reviewer #2 (Remarks to the Author):

The authors performed a thorough revision of the manuscript by substantially modifying the text and adding new experimental results. All of my comments were exhaustively addressed. Most importantly, the authors clearly conveyed the novelty and broad appeal of their study in their response letter. I find the revised manuscript (and the supporting information) to be of excellent quality and befitting publication in Nature Communications as is.

Reviewer #3 (Remarks to the Author):

The authors have adequately addressed all of my concerns, particularly the emphasis on redox control as opposed to the role of electric fields in this study. The control experiments supported their claims on possible reaction mechanisms, and all minor comments are suitably handled. I therefore endorse the publication of this study.

Response Letter to Reviewers' Comments

Reply to Reviewer #1

Comments: The authors have revised the manuscript adequately and have addressed my concerns. I recommend publication.

Reply: We appreciate the reviewer for kindly recommending this work.

Reply to Reviewer #2

Comments: The authors performed a thorough revision of the manuscript by substantially modifying the text and adding new experimental results. All of my comments were exhaustively addressed. Most importantly, the authors clearly conveyed the novelty and broad appeal of their study in their response letter. I find the revised manuscript (and the supporting information) to be of excellent quality and befitting publication in Nature Communications as is.

Reply: We appreciate the reviewer for providing positive feedback on our revision and supporting publication.

Reply to Reviewer #3

Comments: The authors have adequately addressed all of my concerns, particularly the emphasis on redox control as opposed to the role of electric fields in this study. The control experiments supported their claims on possible reaction mechanisms, and all minor comments are suitably handled. I therefore endorse the publication of this study.

Reply: We appreciate the reviewer for the positive feedback and recommendation to publish this work.